# Validation of a New Soccer Shooting Test Based on Speed Radar Measurement and Shooting Accuracy

**DOI:** 10.3390/children10020199

**Published:** 2023-01-20

**Authors:** Felix Engler, Andreas Hohmann, Maximilian Siener

**Affiliations:** 1BaySpo—Bayreuth Center of Sport Science, University of Bayreuth, Universitätsstraße 30, 95447 Bayreuth, Germany; 2Institute of Professional Sport Education and Sport Qualifications, German Sport University Cologne, Am Sportpark Müngersdorf 6, 50933 Cologne, Germany

**Keywords:** soccer, football, talent prognostic, talent identification, shooting test, shooting speed, shooting accuracy, radar measurement

## Abstract

Due to poor results, the German talent identification and development of the German soccer association DFB no longer performs a shooting test since a few years. The aim of this study was to create and validate a new soccer shooting test that allows valid conclusions to be drawn from the shooting quality of youth soccer players about their overall soccer skills. The shooting test was performed with a total of 57 male club players (age: 15.24 ± 0.864 years) from four different teams from the first, second, fifth, and the seventh division of the respective age group (under 15-year-olds until under 17-year-olds). Each subject took one shot at maximum shot speed and eight target shots, measuring accuracy and the shot speed. A multivariable linear regression analysis with forward selection revealed significant values for the variables *average shot speed nondominant leg* (*p* < 0.001) and *total score* (*p* = 0.004; accuracy × speed of every target shot). Based on these two variables, the soccer skills could be derived from the shooting skills of the adolescents in 57.4% of the cases. The study shows the importance of a good technique with the nondominant leg and of the ability to shoot accurately as well as fast simultaneously.

## 1. Introduction

In order for a club or association to establish itself sustainably at national and international levels in the highest, most profitable competitions, it is essential to lay a solid foundation in youth teams. That was one of the reasons for setting up the youth development centers in Germany, by the German Soccer Association (DFB) [1]. Due to the decreasing average age in Germany’s top division [2], the development time for junior players is decreased. Thus, early talent identification and selection (in combination with youth performance development) based on objective criteria plays a decisive role at youth level [3]. Therefore, since 2004, there have been sports scientific performance tests at DFB bases in which all selected players must complete [4]. In order to obtain a comprehensive picture of the overall soccer performance of the base’s players, the following essential skills are tested during the tests: speed, agility, dribbling, ball control, and juggling [5]. The long-term evaluation of the talent development program showed that players with better test results have a relatively better chance of being selected for a further, or higher, development program in the future [6]. However, it is noticeable that an important element, the goal shot, is missing in this test battery. In the past, goal shot was tested at the DFB bases, but the test did not provide reliable results, which is why it was removed from the test battery later.

In order to explain why this happened, it is necessary to take a closer look at the set-up and procedure of the former goal shot test [7]: The subjects played the ball with one touch in a shooting area in front of them. This area was placed centrally in front of the goal at a distance of 14.06 m and was 2.44 m long. After that, the subjects shot the moving ball toward the goal out of motion (the ball had to be shot from a distance of 14.06 m–16.5 m from the goal). The shooting area was 4.88 m wide. The goal was divided into only three areas (left, right, and center); each was 2.44 m in width and height. The subjects were to attempt to shoot at the left and right target fields of the goal with two shots per leg—i.e., to fire a total of eight shots—and to hit the target with their maximum shot speed. Hits were scored as 1 and a miss as 0. Speed was rated only subjectively by the test administrator. A low velocity shot was scored as 1, whereas medium and high velocity shots were scored as 2 and 3, respectively [7]. The test could not fulfill the quality criteria of scientific tests under these specifications. Above all, objectivity is by no means given in the measurement of the shot speed due to the subjective classification of the test administrator into the categories low, medium, and high speed. In addition to this, Höner et al. [8,9] examined the entire test battery of the DFB Talent Development Program regarding its validity and reliability, whereby the goal shot test also turned out to be unreliable (means of Cronbach’s alpha = 0.41; means of test–retest reliability = 0.31).

However, the solution should not be the elimination of the goal scoring test from performance diagnostics because goal scoring is what ultimately decides games. This is illustrated by the relevance of efficiency of chances (i.e., the ratio of goals scored to chances created), which is a decisive factor for winning games [10,11,12]. This, in turn, is conditioned by accuracy and shooting speed, which is why a goal shot test should examine these two variables.

There are already studies that deal with the measurement of shooting accuracy and shooting speed of youth soccer players [13,14,15]. However, these studies focus on the relationship between accuracy and speed of shots and not on performance diagnostics. In addition, the test designs of the mentioned studies used camera systems and evaluation programs, which are not intended for large-scale mass testing. Furthermore, this sensitive data collection is not suitable to be integrated into testing (such as in a DFB base) by untrained test administrators.

The goal of this study is to develop a soccer shooting test that allows us to draw valid conclusions from the shooting quality of youth soccer players about their general soccer quality and performance level. Both the shooting accuracy and the shooting speed are to be considered. These results can later be used to evaluate the shooting skills of players and to identify talents.

## 2. Materials and Methods

### 2.1. General Study Design

In this study, both the shot power based on speed radar measurements and the shot accuracy will be measured. The latter was achieved by dividing a regular men’s soccer goal (7.32 m × 2.44 m, [16]) into five areas (Figure 1). The four areas in the corners of the goal mark the targets at which the subjects should aim. The target net shown in Figure 1, which was made specifically for this study, is attached to the post and crossbar of the goal in an uncomplicated manner using the Velcro tabs shown (Figure 1) and pulled tight. Shot speed was measured using a radar sensor with associated speed indicator from BallSpeedoMeter.de [17]. The radar sensor is placed centrally at behind the goal so that the balls shot at the goal reflect the signal emitted by the sensor back to it, and thus the speed of the balls can be measured.

To be able to make valid statements, teams from different performance classes had to be tested. These included one team from each of the following leagues: 1st division U15 (under 15 years old), 2nd division U17, 5th division U17, and 7th division U15. The measured values were later adjusted for age differences and physical maturity stage during the evaluation to ensure comparable results.

### 2.2. Participants

The soccer shooting test was carried out engaging four different male junior teams of approximately the same age, whereby the performance level of the teams was to vary. Thus, for organizational reasons, the study concentrated on southern Germany, and four teams from different divisions—from the highest to the lowest league—were recruited for the study.

A total of *n* = 57 participants (age: 15.24 ± 0.86 years; height: 174.4 ± 6.9 cm; weight: 59.9 ± 7.4 kg) took part in the study. From the 7th division, 12 subjects (age: 14.01 ± 0.43; height: 170.5 ± 7.2 cm; weight: 55.0 ± 7.1 kg) were tested; from the 5th division, 18 subjects (age: 15.55 ± 0.76 years; height: 175.3 ± 6.2 cm; weight: 62.8 ± 6.0 kg); from the 2nd division, 17 subjects (age: 15.91 ± 0.29 years; height: 178.1 ± 6.2cm; weight: 62.3 ± 7.8 kg); and from the 1st division, 10 subjects (age: 14.99 ± 0.21 years; height: 170.9 ± 5.5 cm; weight: 56.9 ± 6.1 kg) participated in the test.

All participants gave verbal consent for the test to be conducted. The data are available in anonymous form. Ethics confirmation by the university was not applicable.

### 2.3. Test Set-Up

Figure 2 shows the set-up of the goal shot test. The radar sensor (1) is set up in a central position behind the goal at a distance of 4 m from the goal line and connected to the speed indicator of the radar sensor (2). The target net (4) is attached to the crossbar and post of the goal with the Velcro tabs provided (Figure 1) and pulled as tight as possible.

In order to improve the very simplified goal segmentation of the original DFB shot test and thus improve the measurement accuracy of the shot test, the described goal target net (Figure 1) was manufactured especially for this test. The dimensions of the target areas in the corners shown in Figure 1 are given below using the example of the upper left field: The upper side is 3 m long; the left side measures 1.14 m. The lower side, which is the boundary to the target area on the lower left, is 1.5 m long, and the diagonal side on the right is 1.86 m long. These dimensions now result in a target area size of 2.57 m^2^—i.e., less than half the size of the target fields from the DFB’s goal shot test, which are 5.95 m^2^ in size.

The BallSpeedoMeter radar sensor (1) [17] is aligned on a vertical plane to the center of the goal using the height-adjustable tripod on which it is installed. A cable connects the sensor to the associated speed display board (2), which is powered by a rechargeable battery and also supplies power to the sensor. According to the manufacturer, when tested with a set-up such as the one used in this study, the device measures the speed of the ball with a maximum deviation of 1.021%.

Since radar measuring devices cannot be directly standardized, comparative tests with other measuring devices were already carried out in the run-up to the study. Thereby, the speed of a moving object was measured using GPS (Google maps), a laser measurement device (LDM300C by ‘Jenoptic’, Jena, Germany), and a light barrier (Speedtrap 2 by ‘Brower Timing Systems’, Draper, UT, USA) and the BallSpeedoMeter. A correlation analysis proved in our own studies that the results of the BallSpeedoMeter correlated with other measuring devices with a significance of *p* < 0.001 and Pearson’s *r* > 0.983, which shows that the used BallSpeedoMeter has a very high measuring accuracy, making it very suitable for the planned shooting test. Cronbach’s alpha was α = 0.80 for the test–retest reliability of the present sample. Applying the Bland and Altman method [18,19], it can be seen that the second attempt was, on average, 2.81 ± 7.14 km/h (minimum = −15.3, maximum = 20.6) faster than the first attempt.

The test administrator records the results (3) and stands next to the goal so that a good view of both the speed display and the goal is ensured in order to record the results correctly. A cone 5.5 m from the goal line in a central position (5) serves as a marker for the maximum shot speed test. The shooting area from which the subjects have to take the target shots (6) is marked by four cones. It is 4.88 m wide and is located centrally in front of the goal at a distance of 14.06 m. The length is 2.44 m, which then places the cones far from the goal at a distance of 16.5 m from the goal line. Behind the shooting area, there is the starting position of the subjects (7). There, the normal game balls are ready—ideally eight of them, so that the subjects can perform their eight target shots without interruption.

### 2.4. Measurements

The test was conducted following the specified procedure: The subjects warm up collectively as part of their normal practice routine. After warming up, the group begins their normal practice, and in parallel, subjects are taken out of practice one by one to perform the goal shot test.

First, the anthropometric data of the subject were noted. This included date of birth, height, weight, and the subject’s dominant and nondominant shooting leg. Subsequently, the subject shot a stationary ball from the marking of the maximum shot speed test (Figure 2, No. 5) on the goal with maximum shot speed. For this single shot, the speed indicator of the radar sensor (Figure 2, No. 2) was pointed at the subject so that he could see it, thus providing him with additional motivation. For the following target shots, the speed indicator was turned away such that only the test administrator (Figure 2, No. 3) could see it. This was to ensure that the subject could concentrate on the accuracy of the shots and was not distracted by the shooting speed.

For the target shots, the subject played the ball from the starting position (Figure 2, No. 7) with one touch into the shooting area and then shot the moving ball out of the shooting area (Figure 2, No. 6) into the goal. The ball was to be played in motion in order to match the finishing situation as closely as possible to a game situation and not to play stationary balls, as in the case of a penalty or free kick. It was then started with four shots with the dominant leg (DL) in each case, with the targets in the corners to be aimed at in the following order: upper right, lower right, upper left, and lower left. Then four shots were shot with the nondominant leg (NDL) in the same target sequence. The order for this was fixed and identical for every subject, so that in case of accidental hits into wrong targets, the subjects could not claim that they had just aimed there. As a support, the target to be aimed at was briefly called out to the subjects before each shot, in order to avoid misunderstandings. After all nine shots (one maximum speed shot, eight target shots) had been made and noted by the test administrator, the subject was released back into the running practice, and the next subject came to the test set-up.

The accuracy of the tested shots was recorded using accuracy factors. For this purpose, hits were assigned a score of 3, boundary hits (i.e., the ball hit the band or post or crossbar of the desired target) were assigned a score of 2, and misses (i.e., the ball missed the goal or went into another target) were assigned a score of 1. The speed of the shot in km/h was then multiplied with the accuracy factor of the associated shot. Thus, scores were available for each individual shot, which could be added up as desired (depending on the goal of the analysis). For example, the variable *score NDL* describes only the score of the shots with the NDL, whereas the variable *total score* includes all scores of all target shots.

### 2.5. Coach Ranking

A ranking of the tested players on their team was created by each coach after the test. However, this ranking was not influenced by the results of the test, as the results were not yet available to the coaches. In this process, the coaches were asked to rank their players from good to poor based on overall shooting quality. Subsequently, the rankings of the coaches were combined into an overall ranking to serve as a reference value for the analyses later on: The 1st division player ranked best by his coach was assigned first place. The best rated player of the 2nd division was assigned the eleventh place, since 10 players from the 1st division were tested, and consequently, he was the next best player. Thus, the last place in the overall ranking was occupied by the player who was rated the worst by his coach from the worst team (i.e., 7th division). Hence, there was a continuous overall ranking, which was based on the assumption that the worst player from a team still had better shooting qualities than the best shooter from the team of a division below. It should be noted that although *n* = 57 players were tested, only 54 players appeared in the overall ranking, because on the day of the testing, three players from the 2nd division team were present for a tryout, so the coach did not have sufficient knowledge about their abilities. These players participated in the test, but the coach was not able to make a statement about the shooting qualities of these players, which is why they could not be included in the ranking.

### 2.6. Statistical Analyses

All statistical analyses were performed using SPSS (version 26.0; SPSS Inc., Chicago, IL, USA) with a significance level of *p* ≤ 0.05.

To ensure comparability of the data despite the age differences between the teams, the measured values were adjusted for age. For this purpose, linear regression analyses were calculated, each of which placed the corresponding test variable (dependent variable) in relation to age in months (independent variable). From this, z-standardized residuals were then generated. Thus, age-adjusted values for each variable were available for further calculations. All values were approximately normally distributed, as shown by the Shapiro–Wilk test (*p* > 0.05) and a visual inspection of the histograms.

To obtain an overview of the explanatory power of the different test variables, a one-factor analysis of variance (ANOVA) was performed to analyze the differences between the teams or leagues regarding their test results. Afterward, a post hoc test (Bonferroni/Dunnet-T3) was used to test the pairwise differences between the mean scores of the teams.

In addition, a bivariate correlation analysis by Spearman was to show which test variables correlates with the coach ranking.

A regression analysis was necessary to finalize the variables with the best explanatory power. For this purpose, multiple linear regression analyses with forward selection were calculated with the test variables as independent variables and the coach ranking as the dependent variable (missing values: pairwise). The independent test variables included in the calculations were the uncorrected raw scores in a first analysis and the age-corrected values in a second analysis. This was supposed to provide information on whether the age differences of the adolescents have an influence on the results.

## 3. Results

Table 1 shows the descriptive statistics with the test results of the shooting test. We notice that *score lower targets* achieves significantly better values than *score upper targets*. In addition, despite the age differences, clear differences can be seen between the performance levels regarding *total score*. For *max. shot speed*, the two older teams achieved better values. Furthermore, the subjects from the highest performance level were able to shoot the hardest in relation to their maximum shot speed in the target shots, which is shown by the variable *avg. shot speed/max. shot speed*.

### 3.1. Comparison of Performance Levels

The ANOVA performed showed significant group differences between the four teams for almost all test variables. Only for the test value *shots at post* could no statistically significant difference between the teams be shown. The results of the ANOVA are shown in Table 2. The results of the post hoc test was marked as significant differences between the performance levels in Table 1. In contrast to the results of the ANOVA, the post hoc test showed that there were also no significant differences in some other variables. Thus, in addition to *shots at post*, no significant differences between the individual groups could be detected for *shots missed*, *score DL*, and *score upper targets*. Nevertheless, it can be seen that especially the group from the 1st division stands out from the two lower performance levels. There are no significant differences between the 1st division and the 2nd division in any of the variables. The 7th division performed the worst. It had significantly worse scores on most variables (8 of 13) than at least one other team. The teams from the highest and lowest levels differed most frequently (also eight times).

### 3.2. Correlation Analysis

The correlation analysis revealed that, except for *shots at post*, all variables correlated significantly with coach ranking at a significance level of 0.01. The Pearson R-values of the bivariate correlation analysis are shown in Table 3. The fact that *shots at post* is the only variable that does not correlate significantly with coach ranking and reaches only barely significant values overall is consistent with the poor results for this variable in the ANOVA. The remaining values correlate strongly with the coach ranking (*p* < 0.01).

### 3.3. Predictability of Soccer Quality through the Shooting Test

The multivariable linear regression with forward selection showed that only the two raw variables *avg. shot speed NDL* (*p* < 0.001, standardized coefficient = −0.498) and *total score* (*p* = 0.004, standardized coefficient = −0.394) had a significant influence on the model (all other variables were excluded due to nonsignificant influence). Accordingly, based on these two variables, it is possible to draw conclusions from the shooting quality to the soccer quality of the subjects. The analysis reached a corrected *R*^2^ of 0.574 and *p* < 0.001. The regression equation is as follows (Equation (1)):(1)Ranking position=108.103−0.858×avg.speed NDL−0.025×total score

If the same analysis is calculated with the age-adjusted *z*-scores, the same results appear. Again, the two test values from *avg. shot speed NDL* (*p* < 0.027) and *total score* (*p* = 0.014) are the only significant variables. However, the corrected *R*^2^ value decreases to 0.364.

This shows that the two included independent variables are negatively related to the dependent variable, the coach ranking. The negative correlation can be explained by the order of the coach ranking, which indicates the best rank with the lowest number.

Figure 3 and Figure 4 show the ranges of the variables *avg. shot speed NDL* and *total score* in each case in relation to the performance level. The differences between the teams are recognizable and allow areas to be defined on which basis a subject can be assigned to a performance level based on his shot quality. The overlaps show that there are smooth transitions between players of different levels, e.g., it is possible that some players from a lower league achieve better results in the shot test than a player from a higher league and vice versa. The presented mean lines show a clear loss of value for both variables the lower the performance level of the team is. Only the differences in *avg. shot speed NDL* between the 1st division and the 2nd division are not very large. However, the difference in performance level between these two groups is, in fact, the smallest.

## 4. Discussion

The goal of the study was to create a goal shot test that is suitable for making valid statements about the general soccer skills of adolescent soccer players based on the shooting quality by measuring shooting accuracy and shooting speed. Based on the test results of the subjects in interaction with the available classifications of the coaches about their players, the variables *avg. shot speed NDL* and *total score* turned out to be an accurate way to predict performance. The explanation for this is that the *total score* includes both the accuracy and the speed of all shots and thus represents the best overall picture of a player’s qualities. When trying to shoot accurately, shooting power decreases, and, in turn, accuracy suffers when trying to shoot very hard [14,15]. Hence, the difference between the different performance levels of players is not the maximum shot speed but rather whether players can still shoot relatively hard when they are supposed to shoot accurately, or whether they can shoot hard and accurately at the same time. This ability is comprehensively represented by the *total score* variable. The results of this study show that players from the higher performance levels seem to manage to shoot hard and accurately more easily than those from the lower divisions.

In addition, of great significance is the *avg. shot speed NDL*, since at youth age players from lower performance levels train less and can therefore focus less on training their weak foot than in higher performance levels, where practice takes place more frequently and is more intense. Ultimately, this is reflected in the *avg. shot speed NDL*, as poorly trained shooting technique with the NDL results in being able to shoot less strongly with it. The better results on this variable (i.e., better technique of the NDL) of the higher performance levels in this study coincides with a review article by Stoeckel and Carey [20]. They conclude that professional soccer players are not technically worse with the NDL than with the DL due to a lot of training. Transferred to the present study, this means that players of higher quality are technically better with the NDL, which is reflected in *avg. shot speed NDL*, as mentioned before.

Studies by Ali et al. [21] and Keller et al. [22] showed that elite soccer players on average shoot harder than nonelite soccer players. Professional athletes reached an average speed of about 80 km/h in their studies. The average shooting speed of the highest performance group in our sample was 72 km/h. This difference can probably be attributed to the age of the participants—they analyzed adult soccer players in their studies, whereas junior soccer players were tested in our study.

It is important to consider age and developmental effects, especially in the case of unevenly aged subjects, as in this study, because they influence talent identification [23,24,25]. Remarkably, the effects of age differences among teams do not appear to be evident in the results of this study, which contrasts with results of previous studies [26,27,28]. These studies investigated the development of shooting power over the developmental progression of adolescents and showed that shooting power increases the older the player becomes. Therefore, it would have been expected that in this study the adjusted values of the variables would produce better results. However, this was not the case. One explanation for this could be that the age differences were not severe, as in the previous studies. In addition, it is possible that these effects eliminated themselves because the two younger groups were those with the highest and those with the lowest performance levels. For the same reasons, it is possible that this study was unable to show age and developmental differences in terms of shooting accuracy. This also contradicts findings of other studies [13,29,30]. Given that there are no detectable effects, no conclusions can be drawn about the differences between the age and development effects that Towlson et al. [29] confirm.

Studies evaluating penalty kicks show that the probability of scoring is significantly greater when the ball is shot into the top corners of the goal compared to when it is placed at the low [31,32]. However, these studies also show that significantly more shots are taken at the bottom, as the risk of missing is significantly greater when shooting high. Thus, it is significantly more difficult to hit the upper corners, which is consistent with the results of the present study: every team tested scored better on *score lower targets* than on *score upper targets* (Table 1). Yet, the question exists how relevant the placement in the upper or bottom corners of the goal is in open game situations. Pertsukhov and Shalenko [31] found that in the 2019/2020 UEFA Champions League season, 68.1% of all goals were scored by shooting to the lower part of the goal. Consequently, the shot quality of low shots seems to be quite relevant for open play. Related to the results of this study, we can see that the better teams score significantly better on *score lower targets* than the worse teams. In contrast, there are no significant differences for *score upper targets*. Thus, it appears that the probability of scoring is higher for shots at the upper part of the goal, but so is the risk of missing the target entirely. Low shots are more likely to be blocked, but more goals are scored with low shots, which is consistent with the results of this study. Therefore, low shots seem to be more relevant than high shots due to the higher number of shots taken and the more goals that are scored.

It needs to be mentioned, however, that despite good results from the correlation analysis, the results of this study may have been affected by inaccuracies in the coach rankings because they relied solely on the subjective assessment of the coaches. One solution to this problem could have been to have each coach evaluate each player from each team. Yet this was not feasible. Therefore, due to lack of alternatives, it had to be assumed that the worst player of the best team would be ranked even better than the best player of the second worst team. This solution is not ideal, because in practice it is very likely that there is qualitative intermixing or overlapping between the teams, especially when focusing on only one skill (shooting), as can be seen in Figure 3 and Figure 4. Another solution could be a retrospective study after a few years, testing the same subjects in adulthood and then comparing them with the current values and their performance level at the men’s level. The results of the ANOVA performed, which assumes independence of the variables, should be interpreted cautiously. Due to the test design, there may be intercorrelations between variables.

For the future, it is necessary to perform the test with a larger number of adolescents in order to consolidate the results of this study. In doing so, attention should be paid to eliminate or minimize the limitations mentioned above. Talent diagnostics is a complex subject of research, because especially in sports such as soccer, it depends on many different factors that can contribute to a player reaching the professional level in later life [33]. Ultimately, it cannot be the goal to predict the overall soccer skills of a player only based on a specific shooting test. However, especially from this point of view, the results of the study are positive, showing that 57.4% of the soccer performance of youth players given by the coach ranking can be derived from the total score (*total score*) and the average shot speed with the NDL (*avg. shot speed NDL*). Furthermore, players playing in the highest performance level of their age group achieved a total score of 1219.45 points on average in the shooting test and averaged 70.223 km/h shooting with their weak foot. These values can serve as first reference values for further testing among subjects in this age group.

## 5. Conclusions

The study successfully demonstrated that the shooting quality is an indicator for the overall soccer performance of a young player. The slightly modified test, based on the former DFB-shooting test [7] achieved valid results. It was remarkable that the age of the youth soccer players did not seem to have any influence on the results of this study. The test showed that it was possible to draw conclusions about the overall soccer skills based on the shooting quality in far more than half of the cases on the basis of two parameters (*avg. shot speed NDL* and *total score*). Given that soccer is about more than just shooting skills and that there are many other qualities that make a good soccer player, these results should be viewed positively.

## Figures and Tables

**Figure 1 children-10-00199-f001:**
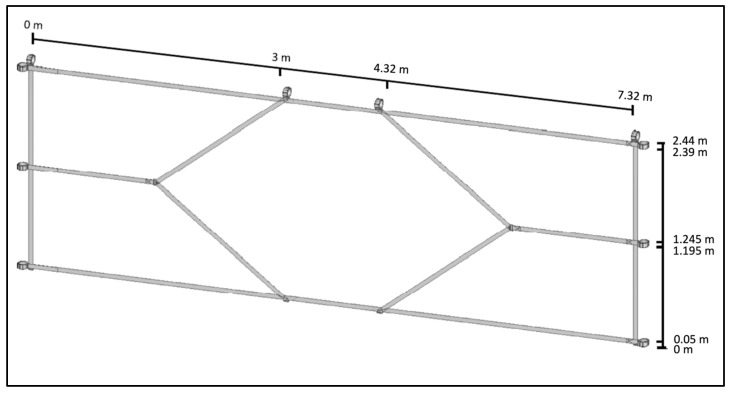
Target net which is attached to the posts and the crossbar of the goal.

**Figure 2 children-10-00199-f002:**
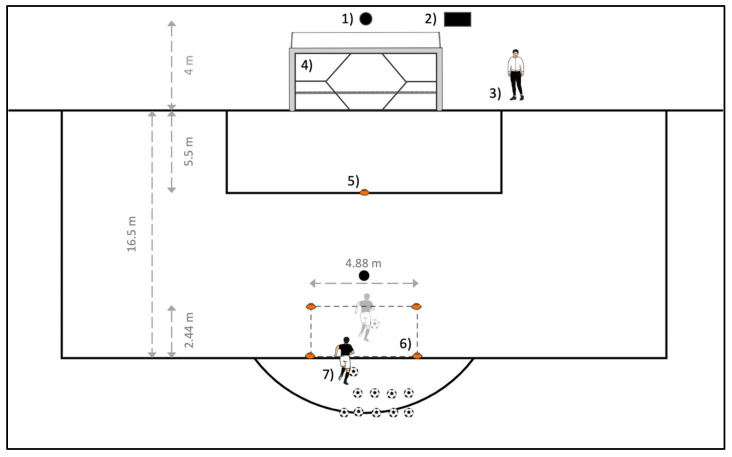
Test set-up for the shooting test. (1) Radar sensor, (2) Speed indicator of the radar sensor, (3) Test administrator with evaluation sheet, (4) Target net, (5) Marker of the maximum shot speed test, (6) Shooting area for target shots, (7) Starting position of the subjects.

**Figure 3 children-10-00199-f003:**
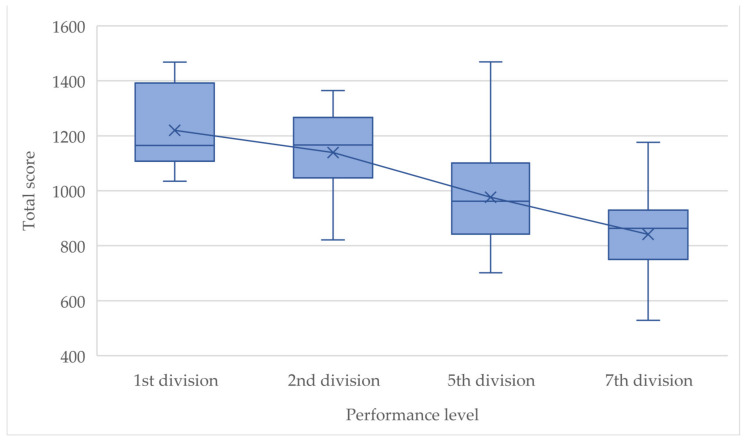
Boxplots of *total score* by performance level.

**Figure 4 children-10-00199-f004:**
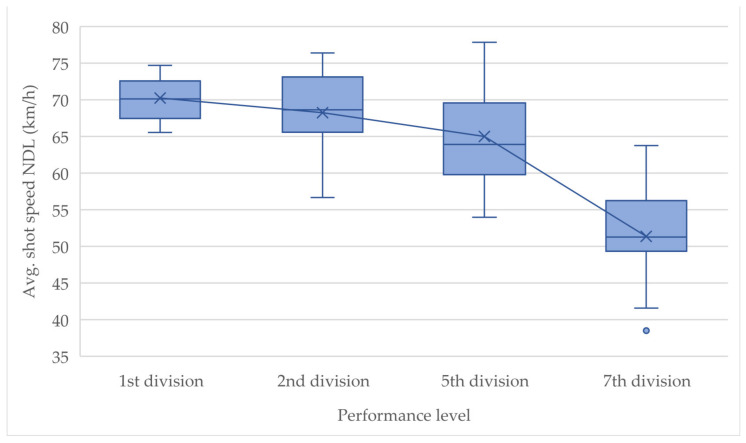
Boxplots of *avg. shot speed NDL* by performance level.

**Table 1 children-10-00199-t001:** Descriptive statistics with noted significant differences between performance levels shown through the post hoc tests.

Variables	Performance Level (Sign. Difference)	N	M	SD	SE	95%-CL		Min	Max
						LL	UL		
**Shots on target**	1st division (c)	10	3.90	0.994	0.314	3.19	4.61	2	5
	2nd division (c)	17	3.59	1.176	0.285	2.98	4.19	1	5
	5th division (a,b)	18	2.50	1.249	0.294	1.88	3.12	1	5
	7th division	12	2.67	1.073	0.310	1.98	3.35	1	5
**Shots at post**	1st division	10	1.00	1.155	0.365	0.17	1.83	0	3
	2nd division	17	0.88	0.928	0.225	0.41	1.36	0	2
	5th division	18	1.00	0.840	0.198	0.58	1.42	0	2
	7th division	12	0.83	0.835	0.241	0.30	1.36	0	2
**Shots missed**	1st division	10	3.10	1.101	0.348	2.31	3.89	1	4
	2nd division	17	3.53	1.281	0.311	2.87	4.19	1	5
	5th division	18	4.50	1.465	0.345	3.77	5.23	2	7
	7th division	12	4.50	1.382	0.399	3.62	5.38	3	7
**Max. shot speed (km/h)**	1st division (d)	10	93.10	6.631	2.097	88.356	97.844	81.2	103.8
	2nd division (d)	17	97.67	8.000	1.940	93.557	101.784	83.7	109.0
	5th division (d)	18	98.12	6.894	1.625	94.700	101.556	84.8	108.1
	7th division (a,b,c)	12	82.45	6.633	1.915	78.235	86.665	68.1	90.7
**Avg. shot speed (km/h)**	1st division (d)	10	72.76	2.584	0.817	70.9130	74.6095	69.08	76.78
	2nd division (d)	17	70.66	5.601	1.359	67.7854	73.5455	60.15	79.38
	5th division (d)	18	70.00	5.446	1.284	67.2938	72.7103	60.40	79.14
	7th division (a,b,c),	12	58.75	4.560	1.316	55.8569	61.6514	48.51	65.73
**Avg. shot speed DL (km/h)**	1st division (d)	10	75.30	2.943	0.931	73.1944	77.4056	70.48	78.85
	2nd division (d)	17	73.10	6.173	1.497	69.9291	76.2768	62.58	82.75
	5th division (d)	18	75.00	6.248	1.473	71.8928	78.1072	65.25	87.85
	7th division (a,b,c)	12	66.13	5.823	1.681	62.4313	69.8312	55.53	74.90
**Avg. shot speed NDL (km/h)**	1st division (d)	10	70.22	3.025	0.957	68.0582	72.3868	65.53	74.70
	2nd division (d)	17	68.23	5.767	1.399	65.2631	71.1928	56.68	76.38
	5th division (d)	18	65.00	7.036	1.658	61.5051	68.5032	53.98	77.83
	7th division (a,b,c)	12	51.38	6.784	1.958	47.0668	55.6874	38.50	63.75
**Avg. shot speed/max shot speed (%)**	1st division (c,d)	10	78.37	3.730	1.180	75.70	81.04	73.81	85.33
	2nd division	17	72.70	7.263	1.762	68.97	76.43	60.82	86.84
	5th division (a)	18	71.43	4.381	1.033	69.25	73.61	62.73	80.18
	7th division (a)	12	71.37	3.846	1.110	68.93	73.81	64.48	77.40
**Total score**	1st division (c,d)	10	1219.5	149.213	47.185	1112.71	1326.19	1035.10	1467.70
	2nd division (c,d)	17	1138.9	168.131	40.778	1052.46	1225.34	821.30	1364.20
	5th division (a,b)	18	976.3	194.484	45.840	879.59	1073.01	701.60	1469.00
	7th division (a,b)	12	841.8	166.780	48.145	735.82	947.76	528.50	1175.80
**Score lower targets**	1st division (c,d)	10	745.93	71.481	22.604	694.80	797.06	658.60	890.10
	2nd division (c,d)	17	728.46	146.018	35.415	653.38	803.53	474.40	909.00
	5th division (a,b)	18	607.34	137.685	32.453	538.88	675.81	407.80	918.90
	7th division (a,b)	12	501.46	117.148	33.818	427.03	575.89	292.60	689.70
**Score upper targets**	1st division	10	473.52	152.273	48.153	364.59	582.45	278.40	691.70
	2nd division	17	410.44	124.838	30.278	346.26	474.63	255.20	786.80
	5th division	18	368.96	91.774	21.631	323.32	414.59	230.00	550.10
	7th division	12	340.33	92.160	26.604	281.78	398.89	228.40	526.20
**Score DL**	1st division	10	626.27	107.000	33.836	549.73	702.81	450.40	791.50
	2nd division	17	616.11	103.513	25.106	562.88	669.33	411.80	777.40
	5th division	18	519.50	116.663	27.498	461.48	577.52	261.00	767.70
	7th division	12	515.08	170.579	49.242	406.69	623.46	266.60	860.70
**Score NDL**	1st division (c,d)	10	593.18	77.928	24.643	537.43	648.93	459.80	694.30
	2nd division (d)	17	522.79	96.162	23.323	473.35	572.24	340.70	659.70
	5th division (a,d)	18	456.80	107.892	25.431	403.15	510.45	256.70	701.30
	7th division (a,b,c)	12	326.72	84.348	24.349	273.12	380.31	192.30	447.20

DL = dominant leg; NDL = nondominant leg; Differs significantly from (a) 1st division (b) 2nd division; (c) 5th division; (d) 7th division.

**Table 2 children-10-00199-t002:** Results of ANOVA.

Variables	*F* (3.53)	*p*	*η* ^2^
**Avg. shot speed NDL**	23.599	<0.001 **	0.572
**Avg. shot speed**	19.641	<0.001 **	0.526
**Score NDL**	16.535	<0.001 **	0.483
**Max. shot speed**	13.959	<0.001 **	0.441
**Total score**	11.391	<0.001 **	0.392
**Score lower targets**	10.124	<0.001 **	0.364
**Avg. shot speed DL**	6.997	<0.001 **	0.284
**Shots on target**	4.829	0.005 **	0.215
**Avg. shot speed/max. shot speed**	4.433	0.007 **	0.201
**Shots missed**	3.602	0.019 *	0.169
**Score DL**	3.188	0.031 *	0.153
**Score upper targets**	2.891	0.044 *	0.141
**Shots at post**	0.112	0.953	0.006

** High significance; * medium significance; DL = dominant leg; NDL = nondominant leg. Effect sizes (*ƞ*^2^): 0.01 ≤ small, 0.06 ≤ medium, 0.14 ≤ strong.

**Table 3 children-10-00199-t003:** Spearman correlation analysis of test variables and coach ranking.

	Coach Ranking	Max. Shot Speed	Avg. Shot Speed	Avg. Shot Speed DL	Avg. Shot Speed NDL	% of Max. Shot Speed	Total Score	Score Lower Targets	Score Upper Targets	Score DL	Score NDL	Shots on Target	Shots at Post	Shots Missed
**Coach ranking**	-	−0.421 **	−0.684 **	−0.462 **	−0.720 **	−0.409 **	−0.666 **	−0.619 **	−0.402 **	−0.427 **	−0.678 **	−0.433 **	−0.014	0.387 **
**Max. shot speed**		-	0.691 **	0.552 **	0.668 **	−0.338*	0.411 **	0.525 **	0.062	0.277 *	0.409 **	0.057	0.185	−0.167
**Avg. shot speed**			-	0.854 **	0.926 **	0.444 **	0.570 **	0.533 **	0.340 **	0.368 **	0.584 **	0.143	−0.027	−0.027
**Avg. shot speed DL**				-	0.594 **	0.420 **	0.337 *	0.262*	0.269 *	0.299 *	0.261	−0.046	−0.080	0.092
**Avg. shot speed NDL**					-	0.381 **	0.637 **	0.634 **	0.330 *	0.352 **	0.713 **	0.255	0.016	−0.236
**% of max. shot speed**						-	0.245	0.057	0.369 **	0.140	0.269 *	0.135	−0.280 *	0.057
**Total score**							-	0.842 **	0.716 **	0.837 **	0.828 **	0.840 **	0.132	−0.828 **
**Score lower targets**								-	0.226	0.662 **	0.740 **	0.705 **	0.040	−0.650 **
**Score upper targets**									-	0.654 **	0.536 **	0.603 **	0.187	−0.653 **
**Score DL**										-	0.385 **	0.728 **	0.131	−0.728 **
**Score NDL**											-	0.669 **	0.088	−0.648 **
**Shots on target**												-	−0.166	−0.781 **
**Shots at post**													-	−0.487 **
**Shots missed**														-

** Correlation is significant at the 0.01 level (2-tailed); * Correlation is significant at the 0.05 level (2-tailed); DL = dominant leg; NDL = nondominant leg.

## Data Availability

The data of the study are accessible on reasonable request.

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
