# Peer review of "Validation of a New Soccer Shooting Test Based on Speed Radar Measurement and Shooting Accuracy"

_children, 2023, doi:10.3390/children10020199_

Round 1
Reviewer 1 Report
This paper presents a thorough and well-designed study on the validation of a new soccer shooting test. The use of speed radar measurement and shooting accuracy as key metrics is an innovative approach that could have significant implications for soccer training and performance evaluation.
The paper is well-organized and well-written. The introduction and literature are adequate. The presented method is well-detailed.
The necessary tests are exposed. The results of the study are clearly presented and supported by relevant statistical analyses. The finding that the new shooting test is a reliable and valid measure of soccer shooting ability is particularly noteworthy.
Overall, this is a well-conducted and valuable study that makes a significant contribution to the field of soccer research.
Author Response
Dear reviewer,
thank you very much for your throughout positive feedback on our manuscript and for taking the time to review it. Because no specific changes were requested, we did not change the manuscript based on your review.
Reviewer 2 Report
Overall, this was a good study to read, with potential value that can be added to the DFB.
- The introduction is strong with a valid introduction and useful information for the reader.
- Lines 143 - 146: Be careful with correlation in the context of reliability. See https://doi.org/10.11613%2FBM.2015.015 for more info on why this can be misleading.
- The method / procedure was well explained
- I have a slight concern with the ANOVA tests, which assume independence. Given that the data is nested (players within clubs, within divisions) and repeated (8 shots per player), the ANOVA might not be appropriate here. Something like a linear mixed model might be more suitable and produce less biased estimates.
- There was no mention if assumptions of the model were checked (e.g. normality of residuals). Looking at Figure 4, it might be okay, however some groups do appear more variable than others. Data transformation might be required.
- Table 3. has "Pearson" correlation in the title. Lines 228 - 229 has this stated as "Spearman."
Author Response
Dear reviewer,
thank you very much for your positive comments and helpful recommendations on the revision of the manuscript. We appreciated your help very much and intended to address all points raised to the best of our knowledge. The entire manuscript has been revised by us accordingly. Please find our alterations below.
Reviewer:
Lines 143 - 146: Be careful with correlation in the context of reliability. See https://doi.org/10.11613%2FBM.2015.015 for more info on why this can be misleading.
Response:
Thank you for this helpful advice. We have added the results of the method of Altman&Bland proposed. L147 – 149:
“Applying the Bland and Altman method [18,19], it can be seen that the second attempt was on average 2.81 ± 7.14 km/h (minimum = −15.3, maximum = 20.6) faster than the first attempt.”
Reviewer:
I have a slight concern with the ANOVA tests, which assume independence. Given that the data is nested (players within clubs, within divisions) and repeated (8 shots per player), the ANOVA might not be appropriate here. Something like a linear mixed model might be more suitable and produce less biased estimates.
Response:
We have calculated different mixed models, which did not lead to informative results. Therefore, we decided to add a paragraph to the limitations section in the discussion. L389 – 392:
“The results of the ANOVA performed, which assumes independence of the variables, should be interpreted cautiously. Due to the test design, there may be intercorrelations between variables.”
Reviewer:
There was no mention if assumptions of the model were checked (e.g. normality of residuals). Looking at Figure 4, it might be okay, however some groups do appear more variable than others. Data transformation might be required.
Response:
We checked and confirmed the normal distribution of the data. A corresponding reference has now been added. L224 – 225:
“All values were approximately normally distributed, as shown by the Shapiro–Wilk test (p > 0.05) and a visual inspection of the histograms.”
Reviewer:
Table 3. has "Pearson" correlation in the title. Lines 228 - 229 has this stated as "Spearman."
Response:
We have corrected the typing error in the title of table 3. L291:
“Spearman”